# Association between Health Problems and Turnover Intention in Shift Work Nurses: Health Problem Clustering

**DOI:** 10.3390/ijerph17124532

**Published:** 2020-06-24

**Authors:** Jison Ki, Jaegeum Ryu, Jihyun Baek, Iksoo Huh, Smi Choi-Kwon

**Affiliations:** 1College of Nursing, Seoul National University, 103, Daehak-ro, Jongno-gu, Seoul 03080, Korea; candy8@snu.ac.kr (J.K.); huhixoo@gmail.com (I.H.); 2The Research Institute of Nursing Science, Seoul National University, 103, Daehak-ro, Jongno-gu, Seoul 03080, Korea; jgryu21@gmail.com (J.R.); jihyunoctober@gmail.com (J.B.)

**Keywords:** nurses, turnover intention, hierarchical clustering, fatigue, sleep

## Abstract

Shift work nurses experience multiple health problems due to irregular shifts and heavy job demands. However, the comorbidity patterns of nurses’ health problems and the association between health problems and turnover intention have rarely been studied. This study aimed to identify and cluster shift work nurses’ health problems and to reveal the associations between health problems and turnover intention. In this cross-sectional study, we analyzed data from 500 nurses who worked at two tertiary hospitals in Seoul, South Korea. Data, including turnover intention and nine types of health issues, were collected between March 2018 and April 2019. Hierarchical clustering and multiple ordinal logistic regressions were used for the data analysis. Among the participants, 22.2% expressed turnover intention and the mean number of health problems was 4.5 (range 0–9). Using multiple ordinal logistic regressions analysis, it was shown that sleep disturbance, depression, fatigue, a gastrointestinal disorder, and leg or foot discomfort as a single health problem significantly increased turnover intention. After clustering the health problems, four clusters were identified and only the neuropsychological cluster—sleep disturbance, fatigue, and depression—significantly increased turnover intention. We propose that health problems within the neuropsychological cluster must receive close attention and be addressed simultaneously to decrease nurse’s turnover intentions.

## 1. Introduction

Nurses often work irregular shifts and bear high physical and psychological job demands that may, in turn, jeopardize their health status. Specifically, shift work may cause a variety of physical and mental health problems [1]. The deterioration of nurses’ health status could not only lead to a decline in their quality of life but could also affect the quality of care provided by them [2]. In addition, health problems may affect nurses’ turnover, which is a serious issue worldwide [3]. The high turnover rate of nurses has led to an increase in both direct and indirect costs in the health system and could further protract the shortage of nurses that has lasted for the past several years [4]. A recent survey of Korean nurses reported that about 10% of shift work nurses cited health problems as their main reason for resigning [5].

Prior studies also show that nurses complained of two or more health problems simultaneously, which may be interrelated [2,6,7,8]. Musculoskeletal pain in nurses has been reported in many studies [9,10], and poor dietary habits due to irregular shift work were reported to cause gastrointestinal disorders [11,12]. Sleep disturbance, which is most frequently reported in studies of shift nurses, could lead to mood disorders, such as depression, both of which lead to chronic fatigue [13,14,15]. Although nurses experience various health problems, there is relatively little research on the relationships between complex health problems in nurses [2]. Moreover, few studies have investigated the relationship between concomitant health problems and turnover intention.

Because the burden may vary depending on the number and the kind of health problems shift work nurses have [16], it may be important to identify specific comorbidity patterns of nurses’ health problems through clustering and determine which clusters most affect turnover intention, where a cluster-that is, a comorbid pattern of health problems—can be defined as a group of concurrent or related health problems that can be distinguished from other clusters [17].

Therefore, the purpose of this study was to first characterize shift work nurses’ health problems. We then determined the pattern of symptom modalities by clustering the health problems through the hierarchical clustering method. Lastly, we identified the impact of health problem clusters on turnover intention.

## 2. Materials and Methods

### 2.1. Study Design and Participants

This cross-sectional study was part of the Shift Work Nurses’ Health and Turnover (SWNHT) study, which is a prospective cohort study designed to investigate the longitudinal relationships between shift work nurses’ health and turnover. It was supported by the National Research Foundation of Korea (NRF) grant funded by the Korea Ministry of Science and Information and Communications Technologies and approved by the Institutional Review Board (IRB) at two tertiary hospitals in Seoul, South Korea. Data collection was performed from March 2018 until April 2020. In the SWNHT study, we recruited 594 female nurses (294 novice nurses who had no exposure to rotating shift work, and 300 nurses with exposure to 8-hour rotating work, including night shifts, for at least 1 month) (Figure 1). Because health problems can vary according to sex [18,19] and the SWNHT study included a survey of nurses’ menstrual and gynecological symptoms, the SWNHT study was limited to female nurses. Data were collected three times for novice nurses: before exposure to shift work (novice registered nurse (NRN) T0, *n* = 294), after six months of work (NRN T1, *n* = 204), and 12 months after T1 (NRN T2, *n* = 204). For experienced registered nurses, data were collected twice: baseline (experienced registered nurse (ERN) T1, *n* = 300) and 12 months after T1 (ERN T2, *n* = 269; see details in Section 2.2. Data Collection).

In this study, we used data collected from October 2018 to January 2019 (NRN T1, *n* = 204) and from March 2018 to May 2018 (ERN T1, *n* = 300) to analyze the association between health problems and turnover intention among shift work nurses. In this analysis, we defined shift work as a combination of day, evening, and night shifts; therefore, we excluded four nurses, including three nurses who worked only daytime hours and one nurse who worked from midday to 8 p.m.

### 2.2. Data Collection

The primary purpose of the SWNHT study was to investigate health problems, presenteeism, and turnover intention in shift work nurses. To enroll novice nurses without shift work experience, we distributed and collected survey envelope packages that included survey instructions, consent forms, and a questionnaire on the third day of their work orientation before ward placement. To enroll experienced shift work nurses, we attached a recruitment notice to the ward bulletin boards, and nurses who wished to participate in the study voluntarily contacted the research team. We maximized voluntary participation by protecting confidentiality, ensuring anonymity, and no hospital-associated researchers took part in the data collection process. We collected the follow-up data through an online survey program; their response rates were 69.4% (NRN T1, NRN T2) and 89.7% (ERN T1). The SWNHT study questionnaire included questions regarding general and job-related characteristics, health-related variables (e.g., dietary habits, menstrual symptoms, exposure to blood and body fluid, sleep, fatigue, depression, physical activity, etc.), occupational stress, presenteeism, and turnover intention. To objectively verify the sleep scale data, we also obtained actigraphy data from the subjects who consented to wear the actigraphy.

### 2.3. Measurements

#### 2.3.1. Demographic and Job-Related Characteristics

The examined demographic characteristics included age (years), education (bachelor’s degree or lower/master’s degree or higher), marital status (single/married), having children (yes/no), and body mass index (kg/m^2^). The examined job-related characteristics included work unit (general ward, intensive care unit, delivery room, and emergency room), months of shift work experience, and the average number of night shifts per month.

#### 2.3.2. Turnover Intention

We measured turnover intention since it is the most predictive measure of actual turnover [20]. In a longitudinal study in Europe, nurses who had turnover intentions were more likely to leave their jobs [21]. In this study, the subjects were asked to choose one of four options (strongly agree, agree, disagree, or strongly disagree) to answer the question: “I plan on staying for the next year” [22].

#### 2.3.3. Health Problems

The nine health problems in this study were selected by two professors at a nursing college and two nurses in a research team, and were based on reviews of the literature about shift work nurses’ health problems [10,18,23,24,25,26,27]. These were (1) upper musculoskeletal pain (including neck, shoulder, and back pain), (2) leg or foot discomfort, (3) sleep disturbance, (4) fatigue, (5) depression, (6) menstrual disorders (including dysmenorrhea and menopause symptoms), (7) gynecological disorders (including disease of the uterus or ovary), (8) headaches (including migraine, dizziness, and chronic headaches), and (9) gastrointestinal disorders (including gastric ulcer, diarrhea, constipation, and stomachache).

Among the nine health problem categories, sleep disturbance, fatigue, and depression were measured using the instruments described below. For the other six health problem categories, the subjects were asked to indicate the health problems they experienced during the last month with “yes” or “no.”

##### Sleep Disturbance

To assess the quality of sleep, we used the Korean version of the Insomnia Severity Index (ISI), which was developed by Morin and translated by the Korean Sleep Research Society. The insomnia severity scale consists of seven questions related to sleep disorders measured on a 5-point scale (0–4 points) for each item. The score ranges from 0 to 28; higher scores indicate a lower quality of sleep. A score above 10 indicates sleep disturbance [28]. The Cronbach’s alpha value of the Korean version of ISI was 0.928 in our study.

##### Fatigue

Fatigue was measured using the Fatigue Severity Scale (FSS). The FSS consists of nine questions about the degree of fatigue during the past week and is scored from 1 (strongly disagree) to 7 (strongly agree). A higher average score indicates higher fatigue. The criterion for fatigue is more than four points on average [29]. The Cronbach’s alpha value of the FSS was 0.917 in our study.

##### Depression

We measured depression using the shortened Center for Epidemiological Studies Depression Scale (CES-D). The shortened CES-D consists of 10 questions about depressive feelings and thoughts during the past week and is scored from 0 (less than 1 day) to 3 (about 5–7 days). Higher total scores indicate more depressive symptoms. A total score of 10 or above indicates depression [30]. The Cronbach’s alpha value of the shortened CES-D was 0.877 in our study.

### 2.4. Statistical Analysis

All analyses were performed using SAS version 9.4 (SAS Institute Inc., Cary, NC, USA) and R Project for Statistical Computing software version 3.4.4 (CRAN, Soule, Korea). We confirmed that there were no missing data. The descriptive statistics (frequency, percentage, mean, and standard deviation) for the demographic characteristics were analyzed. Pearson’s chi-squared test, Fisher’s exact test, and an analysis of variance were used to identify general characteristics associated with turnover intention. Hierarchical clustering was used to group the health problems reported by participants. Hierarchical clustering is a statistical method for grouping objects or variables according to the similarity between clusters using a bottom-up approach. In the field of nursing, this technique has been used mainly for symptom clustering of cancer patients; however, it has recently become more widely used in various studies [31]. The method used for measuring the distance between variables was the squared Euclidean distance and the linkage method used for measuring the distance between clusters was the average linkage. The number of final clusters is usually determined by the researchers by taking into account clinical suitability [32]. Multiple ordinal logistic regressions that included covariates, such as education, marital status, having children, body mass index (kg/m^2^), work unit, months of shift work experience, and the number of night shifts per month, was used to investigate the association of single health problems and clusters of health problems with turnover intention. The four categories of “strongly agree,” “agree,” “disagree,” and “strongly disagree” used for the turnover intention variable satisfied the proportional odds assumption at *p* > 0.050 with the covariates and variables of interest.

### 2.5. Ethical Consideration

This study was approved by the Institutional Review Board (IRB) at Seoul National University Hospital (IRB No. H-1712-094-907) and the Samsung Medical center (IRB No. 2017-12-075-002). After agreeing to participate in the study, all nurse participants signed a consent form and completed the baseline questionnaire.

## 3. Results

### 3.1. Demographic and Job-Related Characteristics

The participants were 500 female nurses working shifts, including night shifts. The nurses’ mean age was 26.7 years (standard deviation (SD) = 4.20), and 19.8% were over 30 years old. There were no differences in demographic and job-related characteristics between the participants in the two tertiary hospitals. Most nurses were single (88.2%) and had no children (94.0%). The average body mass index (BMI) was 20.19 kg/m^2^; 22.8% of the subjects were underweight and only one subject was obese. The shift work length was 35 months on average, which was highly correlated with age (r = 0.92, *p* < 0.001). Therefore, we excluded age from the covariates of the multiple ordinal logistic regressions (Table 1).

One hundred and eleven nurses (22.2%) had a turnover intention and 12 nurses (2.4%) strongly intended to leave. The turnover intention was statistically higher in subjects who were younger (F = 5.70, *p* = 0.001), had no children (χ^2^ = 10.14, *p* = 0.030), had a lower BMI (F = 4.24, *p* = 0.006), and had shorter periods of shift work (F = 6.83, *p* < 0.001).

### 3.2. Prevalence and Association between Single Health Problems and Turnover Intention

The mean number of health problems was 4.5 (range 0–9), with 95.2% (*n* = 476) of participants having more than two health problems. The most frequently reported health problem was upper musculoskeletal pain (82.4%), followed by leg or foot discomfort (67.8%), fatigue (65.0%), and sleep disturbance (62.4%). The associations between single health problems and turnover intention using multiple ordinal logistic regressions are provided in Table 2. Fatigue (odds ratio (OR) = 3.4, 95% confidence interval (CI) = 2.21–5.24), depression (OR = 1.79, 95% CI = 1.22–2.62), leg or foot discomfort (OR = 1.69, 95% CI = 1.12–2.56), sleep disturbance (OR = 1.61, 95% CI = 1.10–2.37), and a gastrointestinal disorder (OR = 1.51, 95% CI = 1.03–2.19) were significantly related to turnover intention.

### 3.3. Hierarchical Clustering of Health Problems

Based on the hierarchical clustering analysis, four clusters were identified (Figure 2): the pain cluster (upper musculoskeletal pain and leg or foot discomfort), the neuropsychological cluster (depression, sleep disturbance, and fatigue), the gynecological cluster (menstrual disorder and gynecological disorder), and the gastrointestinal cluster (headache and gastrointestinal disorder).

### 3.4. Prevalence and Association between Clusters of Health Problems and Turnover Intention

As a result of our multiple ordinal logistic regression analyses, only the neuropsychological cluster (depression, sleep disturbance, and fatigue) was found to be significantly related to turnover intention. In the neuropsychological cluster, if the participant had only one health problem, it did not relate to turnover intention. If the participant experienced two (OR = 3.35, 95% CI = 1.90–5.92) or three (OR = 5.73, 95% CI = 3.17–10.33) health problems in the cluster simultaneously, the odds ratio of the turnover intention increased linearly, which was statistically significant (F = 5.84, *p* < 0.001; Table 3).

## 4. Discussion

We investigated the prevalence of shift work nurses’ health problems and characterized the patterns of symptom modalities by clustering health problems. We then investigated the association of single health problems and clusters of health problems with turnover intentions. We found that most shift work nurses experienced multiple health problems at the same time. We also found that having more than two health problems in the neuropsychological cluster was significantly related to turnover intention. This study was the first to attempt the clustering of nurses’ health problems and explore the relationship between the clusters and turnover intention in shift work nurses.

We found that 22.2% of nurses had turnover intention. In previous studies, turnover intention varied from 4% to 54% [33,34,35]. The first reason for the difference in turnover intention between existing studies and our study could have been the different measurement tools used in each study. While our study asked about future plans regarding turnover, such as “I plan on staying for the next year,” other studies asked how often they thought about turnover in the past [36,37]. Some studies measured turnover intention with various questions, such as whether they were seeking another job or whether they thought about leaving the nursing profession forever [33,38]. The second reason that turnover intention in our study was higher than in previous studies may be due to different hospital environments. The hospitals where our study was performed were tertiary hospitals in Seoul, which had a higher patient severity and higher nurse labor intensity than other hospitals in Korea. Third, we measured turnover intention and not actual turnover, which is reported to be higher than actual turnover rates [21]. In 2018, the annual average nurse turnover rate was 13.9% in Korea [5].

We found that fatigue was common in our subjects, highly related to turnover intention, and had the highest odds ratio (OR = 3.4, 95% CI = 2.21–5.24). Our results were consistent with a previous study that reported a positive correlation between fatigue and turnover intention [39]. Although we could not determine with certainty how long they had suffered from fatigue, it appeared that fatigue was one of the common disabling health problems that lead to turnover intention. Fatigue may exert a direct effect on turnover intention since nurses’ fatigue has been reported to interfere with work efficiency and concentration and increase the risk of medical error and injury [40,41]. Although the direction of causality was not identified, nurses’ fatigue was reported to be related to sleep disturbance, poor health, and depression [18,25].

Not surprisingly, we found that about 50% of nurses complained of fatigue and sleep disturbance at the same time and sleep disturbance was associated with turnover intention as a single health problem (OR = 1.61, 95% CI = 1.10–2.37). Sleep disturbance has received the most attention as a cause of turnover intention among nurses’ health problems [42,43]. Irregular and insufficient sleep time due to shift work may often cause sleep disturbance, which may affect nurses’ physical and mental health [1]. Another finding of interest was that about 28% of nurses had all three interrelated symptoms of fatigue, sleep disturbances, and depression; this was associated with turnover intention as a single health problem (OR = 1.79, 95% CI = 1.22–2.62). Depression in nurses is prevalent in many studies, and in one study, the prevalence of depression among nurses was almost twice as high as in other professions [26,44,45]. Depression may decrease concentration, which reduces the productivity of nursing and affects nurses’ judgment, thus increasing occupational injury and turnover intention [26,43].

Our study revealed that fatigue, sleep disturbance, and depression may play important roles in increasing turnover intention as a cluster and as individual symptoms. Approximately one-third of nurses experienced all three health problems; these findings suggest fatigue, sleep disturbance, and depression in the neuropsychological cluster were correlated with each other. Despite the fact that biological and behavioral mechanisms in the development of depression, fatigue, and sleep disturbances are unknown, several studies have reported that these three health problems are related and co-occur [46,47]. Most importantly, 80% of nurses experienced one or more health problems in the neuropsychological cluster and this cluster was associated with turnover intention. Moreover, their odds ratio of turnover intention increased linearly as the number of health problems increased within this cluster. Future studies should probe the comorbidity of sleep disturbance, depression, and fatigue of shift work nurses and develop comprehensive health promotion to alleviate these three health problems.

We found that having a gastrointestinal disorder was another common health problem, which was consistent with the result of a previous study of 20,000 Korean nurses [12]. This high prevalence of gastrointestinal disorders among shift work nurses may, first, be due to disturbed circadian rhythm. The gastrointestinal system, like sleep, has a circadian rhythm, which controls bowel movement and the secretion of gastric juices [48]. Second, it might be due to irregular meal times and skipped meals [49]. Although not shown in the result, most of the nurses in our study reported eating irregularly (92.8%) and they ate breakfast twice a week, which was lower than the average number of times Korean adults eat breakfast [50]. The most common reason for skipped meals in our study was irregular work times (64.8%). Considering that having a gastrointestinal disorder was common among shift work nurses and was a single health problem that increased turnover intention, special attention needs to be paid to having regular and sufficient mealtimes as much as possible.

In our results, gastrointestinal disorders and headaches formed the gastrointestinal cluster. This connection could be explained by the association between the brain and the stomach through neural, endocrine, and immune pathways and the high prevalence of headaches in patients with a gastrointestinal disorder [51,52]. However, the gastrointestinal cluster was not related to turnover intention. It is possible that headaches, as an individual health problem, had no significant association with turnover intention, which could have decreased the effect of the cluster. Furthermore, we presume that headaches as a single health problem were not shown to be associated with turnover intention because headaches are often easily relieved by medication and may not have been as severe as a gastrointestinal disorder.

Upper musculoskeletal pain, which had the highest prevalence, formed a pain cluster with leg or foot discomfort. Nurses work most of the time in a standing position, walking an average of 8747 steps (4.1 miles) per shift [53], and high physical demands have been associated with musculoskeletal problems in nurses [54]. Additionally, multi-site musculoskeletal pain has been shown to be more common than single-site pain, especially in women [55]. Unexpectedly, this cluster was not related to turnover intention, although leg or foot discomfort was related to turnover intention. This might be because most nurses (82%) suffered upper musculoskeletal pain regardless of turnover intention and, similar to the gastrointestinal cluster, the association of the pain cluster with turnover intention was reduced by the effect of upper musculoskeletal pain. Although the pain cluster did not relate to turnover intention, given that these health problems in the pain cluster had a high prevalence and cause sickness and absence from work and decreased work productivity [24], there is a need to investigate the prevalence of musculoskeletal disorders in nurses by workplace and to provide appropriate prevention and treatment programs.

Although our study provides a new perspective on nurses’ health problems, it has some limitations. First, this study relied on self-report measures of health problems, except for three health problems (sleep disturbance, depression, and fatigue). Second, we surveyed only the presence of health problems, but not the severity; however, as the participants were nurses with medical knowledge, their judgment of the presence of health problems might be more reliable than that of the general public [56], which would partially compensate for the fact that some health problems were not assessed with standardized tools. Third, we could not infer the causal relationship from the cross-sectional design of the study. The fourth significant limitation is that this study did not measure how many nurses actually leave their job; therefore, the findings of our study may not apply to actual turnover, as turnover intention does not always lead to actual turnover. Fifth, the shift work nurses who participated were all female and from two tertiary hospitals in Seoul in Korea. Therefore, the generalizability of the results is limited. Future studies on the comorbidity of sleep disturbance, depression, and fatigue in shift work nurses from various hospitals in various regions, along with the inclusion of male nurses, are recommended.

## 5. Conclusions

In this study, the association of single health problems and clusters of health problems with turnover intention differed. Although fatigue, sleep disturbance, depression, gastrointestinal disorders, and leg or foot discomfort were related to turnover intention as single health problems, after clustering, only the neuropsychological cluster—including fatigue, sleep disturbance, and depression—was related to turnover intention. Given that nurses had more than two health problems and turnover intention increased linearly within the neuropsychological cluster, these problems must receive close attention and be addressed to decrease the nurse turnover rate. Future studies should implement longitudinal research to determine the effect of the neuropsychological cluster on turnover.

## Figures and Tables

**Figure 1 ijerph-17-04532-f001:**
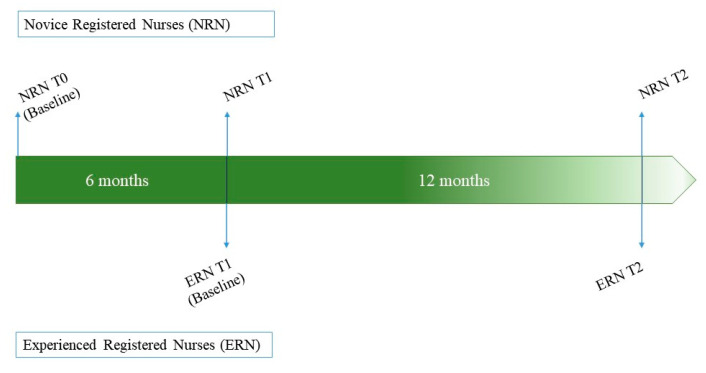
Schematic overview of the Shift Work Nurses’ Health and Turnover (SWNHT) study.

**Figure 2 ijerph-17-04532-f002:**
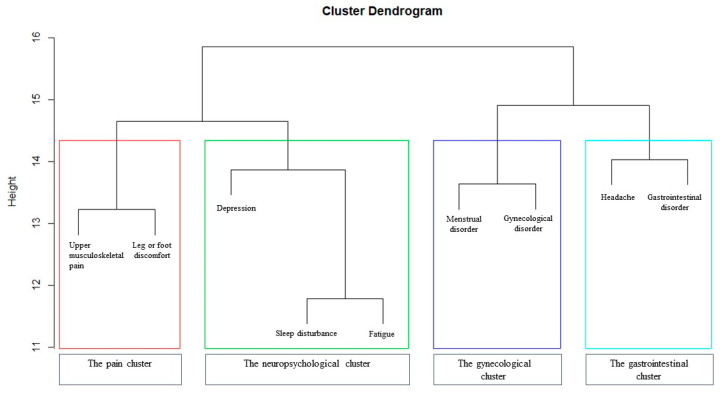
Dendrogram of the health problem clusters.

**Table 1 ijerph-17-04532-t001:** Demographic and job-related characteristics by turnover intention (*n* = 500).

Variables	Categories	Total	Strong Intent to Stay	Intent to Stay	Intent to Leave	Strong Intent to Leave	χ^2^ or F	*p*
		(*n* = 500, 100.0%)	(*n* = 53, 10.5%)	(*n* = 336, 67.2%)	(*n* = 99, 19.9%)	(*n* = 12, 2.4%)		
		*n* (%) or M ± SD	*n* (%) or M ± SD	*n* (%) or M ± SD	*n* (%) or M ± SD	*n* (%) or M ± SD		
Age (years)		26.72 ± 4.20	28.77 ± 5.68	26.29 ± 3.87	27.08 ± 4.31	26.58 ± 2.35	5.70	0.001 *
Education	≤BSN	459 (91.8)	48 (90.6)	310 (92.3)	89 (89.9)	12 (100.0)	1.74	0.626
	≥MSN	41 (8.2)	5 (9.4)	26 (7.7)	10 (10.1)	0 (0.0)		
Marital Status	Single	441 (88.2)	42 (79.3)	303 (90.2)	85 (85.9)	11 (91.7)	6.00	0.108
	Married	59 (11.8)	11 (20.7)	33 (9.8)	14 (14.1)	1 (8.3)		
Having Children	Yes	30 (6.0)	8 (15.1)	15 (4.5)	7 (7.1)	0 (0.0)	10.14	0.030 *
	No	470 (94.0)	45 (84.9)	321 (95.5)	92 (92.9)	12 (100.0)		
Body Mass Index (kg/m^2^)		20.19 ± 2.24	20.96 ± 2.36	20.19 ± 2.19	19.92 ± 2.34	18.77 ± 1.22	4.24	0.006 *
Work Unit	Ward	366 (73.2)	41 (77.4)	239 (71.1)	76 (76.8)	10 (83.3)	4.61	0.673
	ICU	109 (21.8)	8 (15.1)	79 (23.5)	20 (20.2)	2 (16.7)		
	DR, ER	25 (5.0)	4 (7.5)	18 (5.4)	3 (3.0)	0 (0.0)		
Shift Work Experience (months)		34.93 ± 42.94	58.66 ± 59.97	30.84 ± 40.01	37.03 ± 39.38	27.58 ± 27.80	6.83	<0.001 *
Average Number of Nights Per Month (days)		6.00 ± 1.26	5.62 ± 1.48	5.98 ± 1.37	6.06 ± 1.16	6.13 ± 0.78	1.45	0.228

BSN-Bachelor of Science in Nursing, MSN-Master of Science in Nursing, ICU-Intensive Care Unit, DR-delivery room, ER-emergency room; * *p* < 0.05.

**Table 2 ijerph-17-04532-t002:** Association between single health problems and turnover intention using a multiple ordinal logistic regression.

Variables	Total	Adjusted ^1^ Odds Ratio	95% CI	*p*
	(*n* = 500, 100.0%)			
	*n* (%)			
Upper Musculoskeletal Pain	412 (82.4)	1.07	0.65–1.74	0.775
Leg or Foot Discomfort	339 (67.8)	1.69	1.12–2.56	0.012 *
Sleep Disturbance	312 (62.4)	1.61	1.10–2.37	0.013 *
Fatigue	325 (65.0)	3.4	2.21–5.24	<0.001 *
Depression	207 (41.4)	1.79	1.22–2.62	0.002 *
Menstrual Disorder	194 (38.8)	1.26	0.86–1.85	0.229
Gynecological Disorder	36 (7.2)	0.98	0.47–2.01	0.959
Headache	195 (39.0)	1.2	0.82–1.75	0.343
Gastrointestinal Disorder	222 (44.4)	1.51	1.03–2.19	0.031 *

^1^ Adjusted for education, marital status, having children, body mass index (kg/m^2^), work unit, shift work experience (months), and the number of nights per month (days) in multiple ordinal logistic regression model; * *p* < 0.05.

**Table 3 ijerph-17-04532-t003:** Association between clusters of health problems and turnover intention using multiple ordinal logistic regressions.

Cluster	Health Problem			Adjusted ^1^		
	Contents	Number	*n* (%)	Odds Ratio	95% CI	*p*
Pain Cluster	Upper musculoskeletal pain + Leg or foot discomfort	0	37 (7.4)	1.00		
		1	175 (35.0)	0.57	0.27–1.23	0.155
		2	288 (57.6)	1.11	0.54–2.30	0.763
Neuropsychological Cluster	Sleep disturbance + Fatigue + Depression	0	99 (19.8)	1.00		
		1	97 (19.4)	1.59	0.85–2.97	0.141
		2	165 (33.0)	3.35	1.90–5.92	<0.001 *
		3	139 (27.8)	5.73	3.17–10.33	<0.001 *
Gynecological Cluster	Menstrual disorder + Gynecological disorder	0	292 (58.4)	1.00		
		1	186 (37.2)	1.22	0.83–1.81	0.298
		2	22 (4.4)	1.21	0.48–3.05	0.676
Gastrointestinal Cluster	Headache + Gastrointestinal disorder	0	193 (38.6)	1.00		
		1	197 (39.4)	1.43	0.94–2.19	0.092
		2	110 (22.0)	1.60	0.98–2.64	0.060

^1^ Adjusted for education, marital status, having children, body mass index (kg/m^2^), work unit, shift work experience (months), and the number of nights per month (days) in multiple ordinal logistic regression model; * *p* < 0.05.

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
