# Peer review of "Association between Health Problems and Turnover Intention in Shift Work Nurses: Health Problem Clustering"

_ijerph, 2020, doi:10.3390/ijerph17124532_

Round 1

Reviewer 1 Report

Thank you for this excellent study. There are a great many studies on the topic of turnover and the risk factors for it in nursing but this study offers a new perspective and that is the clustering and the outcome is very important for the profession. The unique effect of the neuro-psychological cluster is unexpected. It may be worth mentioning a limitation -common to this kind of work - that the intention to turnover and is not an actual measure of turnover. There are not many studies that do actually measure this but it may be worth mentioning that this could be done to confirm the outcome.

Author Response

Thank you for your valuable feedback. Our responses to your remarks are presented in bold.

It may be worth mentioning a limitation -common to this kind of work - that the intention to turnover and is not an actual measure of turnover. There are not many studies that do actually measure this but it may be worth mentioning that this could be done to confirm the outcome.

Thank you for your suggestion. We have added this to the limitations (lines 304–306).

“The fourth significant limitation is that this study did not measure how many nurses actually leave their job; therefore, the findings of our study may not apply to actual turnover, as turnover intention does not always lead to actual turnover”

Reviewer 2 Report

We suggest decreasing the number of keywords (from 3 to 5). The keywords must reflect the main concepts of the work and appear in the order of their relevance.
The Introduction can be improved in terms of presenting broader evidence on this issue, its impact on the overall health of nurses but also its impact on the health system.
We suggest that the option of only studying female nurses be better clarified.
It is not clear why the headache was included in the gastrointestinal cluster. We suggest better clarification when presenting the results

Author Response

Thank you for your valuable feedback. Our responses to your remarks are presented in bold.

1. We suggest decreasing the number of keywords (from 3 to 5). The keywords must reflect the main concepts of the work and appear in the order of their relevance. 

Thank you for your suggestion. We have decreased the number of keywords to five words that best represent the main premise of our study (line 29).

Keywords: nurses; turnover intention; hierarchical clustering; fatigue; sleep”

2. The Introduction can be improved in terms of presenting broader evidence on this issue, its impact on the overall health of nurses but also its impact on the health system.

Thank you. We have added information to the introduction section to provide a broader perspective of the health of nurses, including its impact on the health system (lines 34–38).

“The deterioration of nurses’ health status could not only lead to a decline of their quality of life, but could also affect the quality of care provided by them [2]. In addition, Health problems may affect nurses’ turnover, which is a serious issue worldwide [3]. The high turnover rate of nurses has led to an increase in both direct and indirect costs in the health system and could further protract the shortage of nurses that has lasted for the past several years [4].”

3. We suggest that the option of only studying female nurses be better clarified.

This study was a part of the Shift Work Nurses’ Health and Turnover (SWNHT) study, which is a prospective cohort study. Another part of the SWNHT study investigated menstrual and gynecological problems among shift work nurses, including premenstrual symptoms measured by the Moos Menstrual Distress Questionnaire. Therefore, the SWNHT study was limited to female nurses.

To clarify why only female nurses were included in this study, we have added to the manuscript, as suggested (lines 67–69).

“Because health problems can vary according to sex [18, 19] and the SWNHT study included a survey of nurses’ menstrual and gynecological symptoms, the SWNHT study was limited to female nurses.”

4. It is not clear why the headache was included in the gastrointestinal cluster. We suggest better clarification when presenting the results

Thank you for pointing out this very important issue. We have further elaborated why the gastrointestinal disorder and the headache formed the gastrointestinal cluster by citing relevant studies (lines 275–283).

“In our results, gastrointestinal disorder and headache formed the gastrointestinal cluster. This connection could be explained by the association between the brain and the stomach through neural, endocrine, and immune pathways and the high prevalence of headaches in patients with a gastrointestinal disorder [51, 52]. However, the gastrointestinal cluster was not related to turnover intention. It is possible that headache, as an individual health problem, had no significant association with turnover intention, which could have decreased the effect of the cluster. Further, we presume that headache as a single health problem were not shown to be associated with turnover intention because headaches are often easily relieved by medication and may not have been as severe as a gastrointestinal disorder.”

Reviewer 3 Report

the manuscript is well-written and the study was conducted appropriately.

Please see line 169 and clarify that the 10.8% is 10.8% of the 111 nurses. The confusion comes from the reported 22.2%, which is of the 500 participants. There should be continuity in the base for the percent report.

the document may be stronger if you propose follow-up research studies based on your findings. This is something that the editor may want to suggest or to leave the conclusion section as is.

Author Response

Thank you for your valuable feedback. Our responses to your remarks are presented in bold.

1. Please see line 169 and clarify that the 10.8% is 10.8% of the 111 nurses. The confusion comes from the reported 22.2%, which is of the 500 participants. There should be continuity in the base for the percent report.

Thank you for pointing this out. We agree with the reviewer’s opinion. We have changed the percentage to be based upon 500 participants to improve understandability for the readers (line 171).

“One hundred and eleven nurses (22.2%) had a turnover intention and 12 nurses (2.4%) strongly intended to leave.”

2. the document may be stronger if you propose follow-up research studies based on your findings. This is something that the editor may want to suggest or to leave the conclusion section as is.

Thank you for your suggestion. We have already mentioned possible future studies in the discussion section (lines 308–310); we have now also added text to the conclusions section (lines 318–320).

“Future studies should implement longitudinal research to determine the effect of the neuropsychological cluster on turnover.”
